# Comparing radiologists' gaze and saliency maps generated by interpretability methods for chest x-rays

**Ricardo Bigolin Lanfredi**
Scientific Computing and Imaging Institute
University of Utah, Salt Lake City, UT, USA
`ricbl@sci.utah.edu`

**Ambuj Arora**
School of Computing
University of Utah, Salt Lake City, UT, USA

**Trafton Drew**
Department of Psychology
University of Utah, Salt Lake City, UT, USA

**Joyce D. Schroeder**
Department of Radiology and Imaging Sciences
University of Utah, Salt Lake City, UT, USA

**Tolga Tasdizen**
Scientific Computing and Imaging Institute
University of Utah, Salt Lake City, UT, USA

## Abstract

We use a dataset of eye-tracking data from five radiologists to compare the regions used by deep learning models for their decisions and the heatmaps representing where radiologists looked. We conduct a class-independent analysis of the saliency maps generated by two methods selected from the literature: Grad-CAM and attention maps from an attention-gated model. For the comparison, we use shuffled metrics, avoiding biases from fixation locations. We achieve scores comparable to an interobserver baseline in one metric, highlighting the potential of saliency maps from Grad-CAM to mimic a radiologist's attention over an image. We also divide the dataset into subsets to evaluate in which cases similarities are higher.

## 1   Introduction

Several interpretability techniques produce saliency maps highlighting the spatial areas of most importance for each of the network's outputs, such as Grad-CAM [19] and spatial attention maps [18]. We propose to quantify[1] the similarity between human attention and the saliency maps produced by these methods. We use the REFLACX dataset [3, 9, 4], which focuses on chest x-rays (CXRs), to build eye-tracking (ET) maps from radiologists' gazes and compare them with saliency maps from abnormality classification models. Figures 1(a) and 1(e) to 1(h) show examples of ET maps and generated saliency maps. Differences can be expected between them. Whereas a radiologist looks at multiple locations to inspect for abnormalities, interpretability methods are expected to highlight the areas where changes would cause a large impact on the output, i.e., abnormalities.

There are reasons for human and model heatmaps to be similar. Grad-CAM, one of the most used explanation methods in the medical field [17], should provide low-resolution smooth saliency maps, similar to the ET maps. When comparing to human heatmaps, Ebrahimpour et al. [7] showed the superiority of the similar class activation map (CAM) [22] method. Since Grad-CAM provides one saliency map per class, we empirically test a few methods of combining them. Attention maps, self-explanatory masks that multiply spatial feature maps of a network, may be similar to ET maps since they are intrinsically class-independent and because of their human attention inspiration [21].

---

[1]Code is available at `https://github.com/ricbl/etsaliencymaps`

We use shuffled metrics to correct for center biases [6]. There is a tendency for fixations, i.e., image locations gazed at by radiologists, to be in central regions of the images. Saliency maps concentrating on these regions achieve high scores independently of image content. Shuffled metrics try to fix this problem and are formulated so that differences and similarities between heatmaps have different weights on the final scores depending on how commonly gazed their locations are. Given the structural similarity of CXRs, we calculate a specific center bias for this task, as shown in Figure 1(d). Finally, we evaluate the generated saliency maps, reaching scores comparable to the interobserver agreement in one of the metrics.

## 1.1 Related work

In the field of interpretability, a few works have used ET maps to evaluate explanatory saliency maps [7, 15, 20]. Ebrahimpour et al. [7] collected ET maps from participants listing objects present in natural images. They compared the data against the saliency maps of the object-detection models for the class with the highest score. Trokielewicz et al. [20] compared the Grad-CAM [19] saliency maps against humans in the task of iris recognition. Muddamsetty et al. [15] did similar work for classification tasks of retinal images. These works did not use metrics to compensate for the biases present in ET data. Karargyris et al. [12] qualitatively checked the Grad-CAM saliency maps against ET maps in CXRs, with no quantitative analysis. To the best of our knowledge, our study is the first to perform this quantitative analysis on CXRs.

The field of automatic generation of ET maps uses ET data as ground-truth and training data [6]. We employ the same comparison metrics as this field, but we do not focus on generating a saliency map that best matches ET maps.

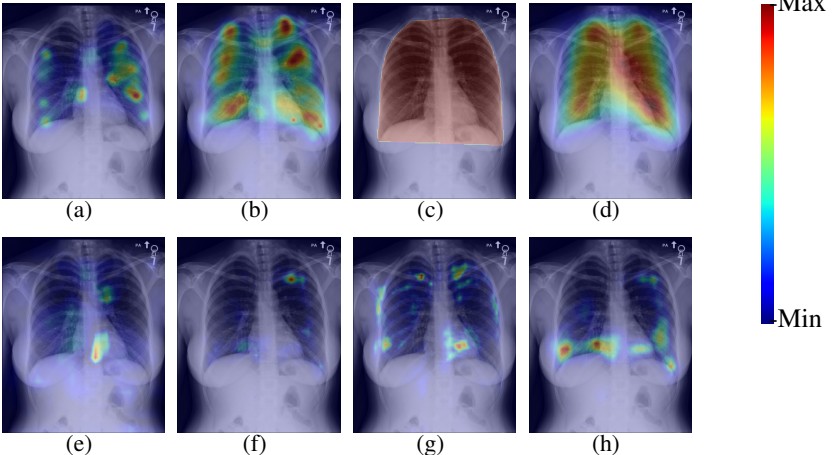

Figure 1: Examples of heatmaps over the respective CXR. a) ET map of a radiologist; b) average ET map of the remaining four radiologists; c) segmentation of the lung region used to calculate a lower boundary for the scores (Section B in the Appendix) d) average ET map from all CXRs, center bias (CB), after registration to match the location of the lungs; e) saliency map for a model without attention gates (woAG) generated by Grad-CAM with uniform weights; f) saliency map for a model with attention gates (wAG) generated by Grad-CAM with uniform weights; g) Attention map 1 (AM1) from wAG; h) Attention map 2 (AM2) from wAG.

## 2 Methods

### 2.1 Grad-CAM

Grad-CAM [19] generates a saliency map according to

$$GC_c = ReLU\left(\sum_k \alpha_c^k LSFM^k\right), \alpha_c^k = GAP\left(\frac{\partial logit(x)_c}{\partial LSFM^k}\right), \tag{1}$$

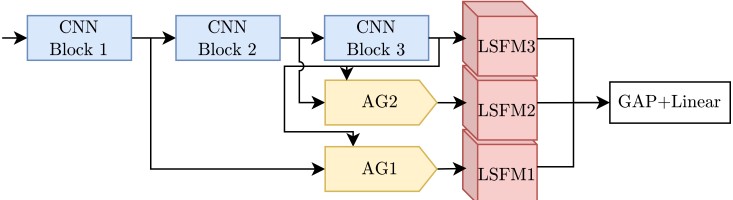

Figure 2: Network used in our experiments. The depiction includes attention gates (wAG). The network without attention gates (woAG) is similar, with only LSFM3 input to the GAP operation. LSFM represents the activation maps used for calculating Grad-CAM. The description of the CNN Blocks is given in Section A from the Appendix, and of the attention gates (AG) in Figure 3.

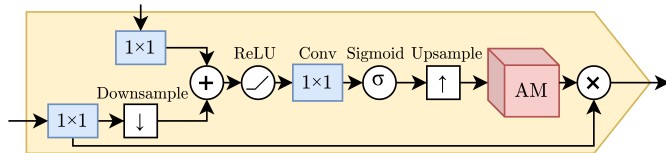

Figure 3: Operations inside the attention gate (AG). AM stands for attention map.

where $GC_c$ is the saliency map for each class $c$, $ReLU$ is a rectified linear unit, $\alpha_c^k$ is a weight for channel $k$ of the last spatial layer of a network, $GAP$ is the global average pooling, $logit(x)_c$ is the logit output for the model being evaluated for class $c$, and $LSFM^k$ are the activations for channel $k$ of the last spatial feature maps of a network. In other words, Grad-CAM calculates a combination of the last spatial feature maps (LSFMs) of a network and the gradient of the network outputs with respect to each element of the LSFMs. To combine the $GC_c$ from all classes, we use

$$\frac{1}{\sum_c \psi_c} \sum_c \psi_c \times GC_c, \tag{2}$$

where $\psi_c$ is a weight for the saliency map of each class. We consider three ways of choosing the weights $\psi_c$ to mix the $GC_c$ for each class $c$:

- **Thresholded**: include classes based on a threshold on the model's output, according to

$$\psi_c = \begin{cases} 1, \text{if } logit(x)_c > 0 \\ 0, \text{if } logit(x)_c < 0 \end{cases} . \tag{3}$$

  If all $\psi_c$ are 0 for an image, we assign $\psi_c = 1$ for the "No Finding" label.

- **Weighted**: weight the classes using the output of the model, according to $\psi_c = \sigma(y_c)$, where $\sigma$ is the sigmoid function.

- **Uniform**: uniformly mix all the classes: $\psi_c = 1$.

## 2.2 Attention gates

Figure 2 shows the architecture of the gated convolutional neural network (CNN), including the location of the attention gates. Figure 3 shows the employed attention gates. Each attention gate provides a saliency map through its attention map, and the attention maps can also be combined into a single saliency map through the use of Grad-CAM.

## 3 Experiments

Experiments employed CXRs and classification labels from the MIMIC-CXR-JPG dataset [9, 10, 11] and ET data from the REFLACX dataset [3, 9]. For each method, we trained five models to calculate the variability of results, which are reported with their 95% confidence intervals. We trained two types of models, with (wAG) and without attention gates (woAG). From the literature on automatic

generation of human saliency maps [6], we selected two metrics to calculate the similarity between human attention and produced saliency maps: normalized cross-correlation (NCC), i.e., Pearson's correlation coefficient, and the Borji formulation of area under the receiver operating characteristic curve (AUC) [5]. We use the shuffled versions of these metrics [6], sAUC and sNCC, to avoid high scores for trivial solutions, such as the highlight of the whole lung. More details about this choice and the shuffled metrics are given in Section B in the Appendix. To reduce the variability of the scores [1] and use the fact that CXRs from the REFLACX dataset had ET maps for five radiologists, we calculated the metrics against the average ET map of all combinations of four radiologists (1 vs. 4). More details about the experiments and networks used can be found in Section A in the Appendix.

## 3.1 Results and discussion

Table 1 reports the metrics for all the tested methods. The slightly positive value of the sNCC metric shows that the ET map from each radiologist is slightly more similar to the average of the other radiologists than to the center bias. Not considering the interobserver baseline, the wAG model using Grad-CAM with uniform $\psi_c$ had the highest scores. Uniform $\psi_c$ might have achieved the best results because radiologists have to look for all abnormalities, including those not found in a particular image. Considering the sNCC metric, one of the models reached scores almost identical to the interobserver evaluation. For the sAUC metric, the interobserver evaluations had the highest scores with a good margin, highlighting that each metric measures different qualities of the heatmaps. Although the attention maps were not the highest scoring saliency maps, the Grad-CAM method had the highest shuffled scores when applied to the wAG model, showing an advantage of attention-gated models when compared to human attention.

Table 1: Shuffled scores for the methods of generating saliency maps (SM). We highlight in bold the highest scores for each metric, excluding the interobserver upper bound. Confidence intervals were calculated using $n = 91$ CXRs. For both metrics, a higher value represents a better result.

| SM | METHOD | sNCC | sAUC |
|---|---|---|---|
| INTEROBS. | BASELINE | 0.028 [ 0.002, 0.055] | 0.558 [0.547,0.568] |
| G-CAM (woAG) | THRESH. | -0.035 [-0.058,-0.012] | 0.510 [0.501,0.519] |
| | WEIGHTED | -0.060 [-0.082,-0.038] | 0.521 [0.511,0.530] |
| | UNIFORM | -0.067 [-0.089,-0.046] | 0.522 [0.513,0.532] |
| G-CAM (wAG) | THRESH. | -0.002 [-0.024, 0.019] | 0.512 [0.504,0.519] |
| | WEIGHTED | 0.027 [ 0.002, 0.053] | 0.528 [0.519,0.537] |
| | UNIFORM | **0.029 [ 0.002, 0.055]** | **0.529 [0.521, 0.538]** |
| AM1 | - | -0.032 [-0.055,-0.009] | 0.514 [0.505,0.523] |
| AM2 | - | -0.007 [-0.034, 0.020] | 0.522 [0.511,0.533] |

We also analyzed scores splitting normal and abnormal cases. Abnormal cases were defined as having a majority of radiologists selecting at least one abnormality for the image. As shown in Table 2, interobserver scores and the scores from a chosen interpretability method were higher for abnormal CXRs. This difference might have been caused by normal cases not having an evident area of interest and abnormalities being areas of longer fixations by radiologists and stronger saliency for Grad-CAM.

Table 2: Scores of baselines and the Grad-CAM (wAG) with uniform $\psi_c$ method when splitting the dataset into normal (N) and abnormal (Abn) CXRs. Confidence intervals were calculated using $n = 17$ normal CXRs and $n = 74$ abnormal CXRs.

| METRIC | LBL | INTEROBSERVER (IO) | GRAD-CAM (WAG) |
|---|---|---|---|
| sNCC | N | -0.056 [-0.110,-0.003] | -0.047 [-0.096,0.002] |
| sNCC | ABN | 0.048 [ 0.020, 0.076] | 0.045 [ 0.016,0.074] |
| sAUC | N | 0.532 [ 0.511, 0.553] | 0.497 [ 0.482,0.512] |
| sAUC | ABN | 0.566 [ 0.554, 0.577] | 0.538 [ 0.529,0.548] |

# 4 Conclusion

Using a dataset of ET data from five radiologists, we showed that, when controlling for center bias, interpretability maps can be as similar to the ET maps from radiologists as ET maps from other radiologists. In other words, although the tested saliency maps might not be good at highlighting areas fixated regularly in the average CXR, they excel at highlighting the specific areas in each CXR that radiologists fixate more than average. In our evaluation, good results were achieved when both Grad-CAM and attention mechanisms were combined, showing an advantage of human-inspired spatial attention gates for generating heatmaps similar to human attention. Moreover, higher similarity scores were associated with the presence of abnormalities, which is similar to our expectations, given that spatial heatmaps are usually formulated to highlight the presence of a class.

## Acknowledgments and Disclosure of Funding

Research reported in this publication was supported by the National Institute of Biomedical Imaging and Bioengineering of the National Institutes of Health under Award Number R21EB028367. The authors declare no competing interests. Christine Pickett provided copyediting support. Vivek Srikumar and Shireen Elhabian participated in discussions related to the project.

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

## A   Experiment details

All CXRs were from patients admitted to the ER at the Beth Israel Deaconess Medical Center from 2011 to 2016. The use of the datasets was exempt from approval since they are publicly available and de-identified. All experiments used a CNN as depicted in Figure 2. We used the parts of the REFLACX dataset where the same CXRs contained ET data for all five radiologists, totaling 91 CXRs. For each of the readings, one heatmap was generated from the fixations. ET maps were generated by drawing Gaussians centered in each fixation and combining them through a sum weighted by the fixation duration. Following Le Meur & Baccino [14], the Gaussians had a standard deviation of 1 degree of visual angle in each axis to represent location uncertainties. We used PyTorch 1.8.1 [16] in our experiments.

Models were trained and validated using the MIMIC-CXR-JPG dataset. We used the Adam optimizer [13] with a learning rate of 0.0001 and a weight decay of 0.00001. After three epochs without improvement on the validation average AUC, we multiplied the learning rate by 0.5. We trained with the binary cross-entropy loss for 75 epochs and a batch size of 64. The 14 labels and the data split from the MIMIC-CXR-JPG dataset were used. All images from subjects displayed to radiologists were moved to the test set. The training was limited to images filtered as follows: images without

classification labels were discarded; only frontal CXRs were kept, i.e., images with "ViewPosition" equals to "AP" (anterior-posterior) or "PA" (posterior-anterior); and studies with more than one frontal image were excluded. The CNN blocks from Figure 2 were built following the Sononet-16 [2] architecture, but with the attention gate inclusion from Schlemper et al. [18]. Each channel from the LSFMs on Figure 2 is considered as one of the $k$ channels from Equation 1. The output from CNN Block 3 in Figure 2 is directed to the attention gates before being called LSFM3 so that the gradient calculation from Equation 1 is influenced by LSFM3 in only one of the three CNN branches.

For training, data were resized to have its shortest dimension equal to 224 pixels, rotated between -15 and +15 degrees, translated by up to 5% of its dimensions, scaled with a scale factor between 0.95 and 1.05, center cropped, randomly horizontally flipped, and normalized by the average intensities and standard deviation of the ImageNet dataset. For validation, images were scaled such that their longest dimension was a multiple of 16, and their shortest dimension was the closest to 224 while keeping the aspect ratio. The image was then padded to a square. Saliency maps were generated with this padded version of the image and then cropped to the original aspect ratio.

On the classification test set ($n = 5$ seeds), woAG had an AUC, averaged over the 14 labels, of 0.774 [0.771,0.776], whereas wAG had an average AUC of 0.769 [0.765,0.772].

## B  The choice of shuffled metrics

### B.1  Details about the original metrics

The NCC was directly calculated between ET map and generated saliency map. The NCC metric calculates the similarity of the saliency maps on the whole image by calculating the correlation between both signals, following

$$\text{NCC}\,(x_1, x_2) = \frac{1}{P-1} \sum \frac{(x_1\,(p) - \mu_{x_1})}{\sigma_{x_1}} \times \frac{(x_2\,(p) - \mu_{x_2})}{\sigma_{x_2}}, \tag{4}$$

where $x_1$ and $x_2$ are the two images of same size, $x_i(p)$ is the value of image $x_i$ in pixel coordinate $p$, $P$ is the total number of pixels in one image, $\mu_{x_i}$ is the average value of $x_i$, and $\sigma_{x_i}$ is the unbiased standard deviation of $x_i$.

The AUC metric considers each pixel of a saliency map to represent the output of a binary classifier deciding if each pixel should contain human fixations or not [6]. This binary classifier should have higher output values for locations associated with fixations from humans (positive locations — ET map) when compared to the rest of the locations (negative locations — uniform heatmap). We calculated the AUC by sampling locations from the heatmaps, which represent spatial probability distributions. The AUC metric calculates the probability of the binary classifier outputting a higher value for a positive location than for a negative location. By looking at the standard deviation of AUCs calculated with a varied number of samples, we found that sampling 1000 positive and 1000 negative samples was enough to have a sufficiently accurate AUC.

### B.2  Sanity check and the use of shuffled metrics

To understand what scores to expect from our experiments, we calculated a higher and a lower boundary for the scores. To have an upper bound for the metrics, we measured interobserver scores. To find a lower bound using a simple algorithm, we segmented the lungs and calculated the convex hull of the segmentations to include the mediastinum and the bilateral hemidiaphragms, as shown in Figure 1(c). Scores for the baselines are presented in Table 3. Upper and lower bounds were practically the same for NCC. This small range was probably caused by a strong bias toward having fixations around the lung area. Thus, using these metrics would not provide much information about our generated saliency maps other than their capacity to highlight the lungs. To correct this bias, we used shuffled metrics [6], which penalize models that output this center bias and reward models that can highlight other regions of the image that were fixated more than the average center bias [6]. We used bounding box annotations for lungs and heart to calculate the center bias in our dataset. We calculated the average bounding box, registered all bounding boxes to the average, applied the same transformation to the ET map, and combined them to get an average of the fixations. The resulting center bias is shown in Figure 1(d). For use in the metrics, the center bias is transformed to match the

bounding box location of the respective CXR. CXRs are very similar regarding what is structurally present in an image, making our center bias issue worse than natural images. We also corrected for the lung location variance, making the similarity with a center bias even stronger. Therefore, the impact of using shuffled metrics for CXR images is probably higher than for natural images.

Table 3: Scores for the upper and lower bounds of the expected results from our methods.

| SM | METHOD | NCC | AUC |
|---|---|---|---|
| INTEROBS. | BASELINE | 0.632 [0.606,0.658] | 0.790 [0.782,0.799] |
| SEGMENT. | BASELINE | 0.637 [0.615, 0.659] | 0.735 [0.726, 0.745] |

For the sNCC calculation, we drew from a closely related metric [6], normalized scanpath saliency, from the shuffled formulation by Gide & Karam [8],

$$sNCC(GT, SM) = NCC(GT, SM) - NCC(CB, SM),$$

where sNCC is the shuffled NCC, GT is the ground truth saliency map, SM is the saliency map being evaluated, and CB is a heatmap representing the center bias in the dataset.

The only change from AUC to sAUC is in sampling negative locations from the center bias map instead of sampling them from the uniform heatmap. This change compensates for the center bias in the metric and assigns higher scores for models that focus on outputting saliency maps highlighting locations that are fixated more than average.

There are differences between the employed metrics. The AUC metric does not penalize false positives, not penalizing saliency maps that are blurrier than the ground truth [6]. The NCC metric is equally affected by false negatives and false positives. sAUC can be between 0 and 1, the higher, the better, and 0.5 represents a model with random outputs. sNCC can be between -2 and 2, the higher, the better, and 0 represents a model with random outputs.

The more extensive range between baseline bounds from Table 4 shows that considering the center bias is essential for calculating a meaningful score.

Table 4: Shuffled scores for the upper and lower bounds of the expected results from our methods.

| SM | METHOD | sNCC | sAUC |
|---|---|---|---|
| INTEROBS. | BASELINE | 0.028 [ 0.002, 0.055] | 0.558 [0.547,0.568] |
| SEGMENT. | BASELINE | -0.187 [-0.204,-0.169] | 0.505 [0.500,0.510] |

