# OpenReview forum: "Comparing radiologists' gaze and saliency maps generated by interpretability methods for chest x-rays"
_NeurIPS.cc/2022/Workshop/GMML — Gaze Meets ML 2022 Poster_

### Official Review · Reviewer_Wf4t · 2022-10-17
**A very interesting idea to see the correlation between radiologists' gaze when analyzing chest x-rays and heatmaps generated by deep learning models.**

**Rating:** 9
**Confidence:** 5

**Review:**

The author(s) of this manuscript propose a very interesting topic, namely to see how heatmaps generated by deep networks agree with locations where human radiologist looks like to identify. This type of research is very useful to somehow explain the current deep architectures by assigning a useful meaning to it. The paper can be identified as an attempt toward interpretable machine learning. The experiment setup is quite thorough and well described (see Appendix A and Appendix B). It is arguable why these details do not appear in the paper itself instead of providing them as "side notes" (see Appendix). Probably it is just a matter of taste.

Despite the fact that the literature in this exact topic is limited, the authors are encouraged to present this research more in-depth, draw some conclusions and explain why their method is better and what gaps they are trying to fill. I think this part is missing from the article.

The description of the methods (see Grad-CAM and Attention gates) is pretty straightforward and very clear. The data considered in the experiment is appropriate and the 5 radiologists also were involved with their own reading. While the authors consider some 91 chest x-rays, calculating single values, it would be interesting to see the same numbers for each pathology separately. This might reveal some pathologies where humans and machines agree and some where they completely disagree, etc.

The results are rather promising, and definitely this research is worth appreciated. We also encourage the authors to publish their data so others can also use it in the future.

P.S.: It is not clear that the authors mention that the manuscript was sent to NeurIPS, so how it ended up here too in this workshop?

---

### Official Review · Reviewer_tiKr · 2022-10-19
**Analysis of gaze data for radiologists**

**Rating:** 7
**Confidence:** 5

**Review:**

Very interesting paper that compares gaze of 5 radiologists with heatmaps such as GradCam.
This can work well on 2D data where the screen is static such as xrays but for 3D imaging where slices change or histopathology where scale is changed this could become a challenge.
Purely technical novelty is maybe limited but I really like the idea. Most base techniques have existed and been used many times before.
I would personally separate the objective results from the subjective discussion, as is usually good practice. I would also not over-interprete absolute results and rather look at the big lines.
The text is well written and easy to read.

---

### Official Review · Reviewer_FZ2U · 2022-10-19
**Interesting work, paper is well written**

**Rating:** 6
**Confidence:** 3

**Review:**

Good overview of background and related work.

The authors are presenting an interesting task, comparing eye-tracking maps with saliency maps.

It's not exactly clear how those 2 compare just by looking at the qualitative results (Figure 1).

---

### Meta-Review · Area_Chair_p7Bo · 2022-10-20

**Recommendation:** Accept (Poster)
**Confidence:** 5

**Metareview:**

The authors present quantitative comparisons of eye-gaze tracking on radiology data and the saliency maps. This preliminary abstract extends the qualitative comparisons of earlier works and attempts to quantify the comparisons between the human gaze and saliency maps. They use shuffled AUCs and Normalized cross-corelation as metrics.

I am recommending an acceptance, which will make for an interesting discussion in the workshop.

---

### Decision · Program_Chairs · 2022-10-20

Accept (Poster)